# Deductive Reasoning and Working Memory Skills in Individuals with Blindness

**DOI:** 10.3390/s22052062

**Published:** 2022-03-07

**Authors:** Eyal Heled, Noa Elul, Maurice Ptito, Daniel-Robert Chebat

**Affiliations:** 1Department of Psychology, Ariel University, Ariel 40700, Israel; danielc@ariel.ac.il; 2Neurological Rehabilitation Department, Sheba Medical Center, Ramat Gan 52621, Israel; noaelul10@gmail.com; 3Navigation and Accessibility Research Center (NARCA), Ariel University, Ariel 40700, Israel; 4School of Optometry, University of Montreal, Montreal, QC H3T 1P1, Canada; maurice.ptito@umontreal.ca; 5Department of Neuroscience, Copenhagen University, 2200 Copenhagen, Denmark; 6Department of Neurology and Neurosurgery, Montreal Neurological Institute, McGill University, Montreal, QC H3A 2B4, Canada; 7Visual and Cognitive Neuroscience Laboratory (VCN Lab), Department of Psychology, Faculty of Social Sciences and Humanities, Ariel University, Ariel 40700, Israel

**Keywords:** congenital blindness, late blindness, deductive reasoning, working memory, executive functions

## Abstract

Deductive reasoning and working memory are integral parts of executive functioning and are important skills for blind people in everyday life. Despite the importance of these skills, the influence of visual experience on reasoning and working memory skills, as well as on the relationship between these, is unknown. In this study, fifteen participants with congenital blindness (CB), fifteen with late blindness (LB), fifteen sighted blindfolded controls (SbfC), and fifteen sighted participants performed two tasks of deductive reasoning and two of working memory. We found that while the CB and LB participants did not differ in their deductive reasoning abilities, the CB group performed worse than the sighted controls, and the LB group performed better than the SbfC group. Those with CB outperformed all the other groups in both of the working memory tests. Working memory is associated with deductive reasoning in all three visually impaired groups, but not in the sighted group. These findings suggest that deductive reasoning is not a uniform skill, and that it is associated with visual impairment onset, the level of reasoning difficulty, and the degree of working memory load.

## 1. Introduction

Executive functions are essential for regulating and organizing our thoughts and actions for the purpose of goal-directed behavior in everyday life [1,2,3]. Working memory [1,3] and deductive reasoning [4,5] are two of the executive functions’ core high-level cognitive abilities. Working memory is responsible for the active maintenance and manipulation of information in the short term [6]. Reasoning is defined as the ability to make inferences, manipulations, and alterations of information. Deductive reasoning is a specific type of higher reasoning skill using verbal, visual, or numeric premises to produce a logical conclusion [7].

Deductive reasoning and working memory are associated with one another in people who are sighted [7,8,9,10,11], and these skills are dependent on perceptual aspects [4], but the exact influence of visual experience on these skills remains unclear. Two opposing theories predict different cognitive performance subserved by neural plasticity in individuals with blindness. The first, the “sensory compensation hypothesis”, suggests that damage in a certain modality may lead to superior performance on cognitive tasks in the intact remaining modalities due to plastic reorganizational changes [12]; the second, the “perceptual deficit hypothesis”, assumes that damage to one sense leads to impairment in the other senses [13].

Research on deductive reasoning and working memory in people who are blind does not clearly support either of these [13,14,15]. For instance, in terms of working memory tested via the auditory modality, certain studies indicate better performance by people who are blind [16,17,18,19], while others find they perform equally well compared to sighted controls [20,21,22]. Working memory can also be tested via the tactile modality, and here too there exist inconsistencies in the literature, as certain studies show superior or equal performance by people with blindness compared to sighted controls [17,22,23,24], while others show deficiencies [22,25,26,27]. In terms of deductive reasoning, the same contradictions exist since certain findings indicate that blind individuals perform worst on deductive reasoning tasks compared to sighted controls [28,29,30], yet others indicate that they perform equally well or better than controls [31,32,33,34,35].

Visual deprivation from birth affects the types of strategies employed [36] and may even confer supranormal abilities to people who are CB in terms of working memory [37], making them immune to certain impedance effects of irrelevant visual information for mental imagery tasks [28]. Comparing these abilities between individuals with CB to those with late-onset blindness (LB) has yielded inconsistent findings [31,33,35,38,39,40]. Two factors can account for this advantage. The first stems from neurological processes at the structural and functional levels that occur as a result of congenital blindness and the second involves mechanisms of training induced brain plasticity (for a review, see [15]), suggesting that intensive use of working memory and enhanced practice leads to improvements compared to those who have not used it as much or rely on other functions for performance [23]. Furthermore, despite the fact that both working memory and deductive reasoning are crucial skills for people who are blind, this relationship is not fully understood in blindness.

This study explores differences between CB, LB, and sighted participants in terms of their working memory and reasoning skills, as well as the relationship between these two skills in groups with varying degrees of visual experience. Two different measures for working memory and for reasoning skills are used in this study: working memory is measured using both the letter–number sequencing task and the digit span backward task, while the deductive reasoning tests are used to evaluate reasoning skills via audition (see Section 2). For each group, the scores for each test and each item are calculated and the relationship between deductive reasoning and verbal working memory is assessed. Our general findings show that vision is not essential for the development of deductive reasoning or working memory skills. Importantly, we also find that deductive reasoning skills are predicted by working memory skills. Psychologists and other professionals in the field of cognitive assessment in educational and vocational settings could consider these important aspects and pay more attention to the relationship between onset of blindness and the abilities that are being measured.

## 2. Materials and Methods

### 2.1. Participants and Ethics

Sixty native Hebrew speakers (30 men), aged 20–43 (*M* = 29.41, *SD* = 6.19) and possessing between 12 and 24 (*M* = 14.46, *SD* = 2.08) years of education were included in the study. The sample is composed of sighted, CB, and LB individuals, separated into four groups. A total of 30 blind participants with no residual vision or light perception recruited from the Center for the Blind in Israel were divided into two groups: 15 CB participants (9 men) and 15 LB participants (8 men). The average age of blindness onset in the LB group was 19.66 (*SD* = 8.65). Additionally, 30 sighted controls with normal vision were divided into two groups: 15 blindfolded participants and 15 using full vision for the task. In order to control for the load on working memory during the reasoning task, a sighted full vision control group performed the task using a pen and paper (see Table 1 for demographics). Using a pen and paper reduces the load on working memory for this task. We excluded participants based on any of the following: (a) developmental disorders with a potential effect on cognitive functioning; (b) diagnosis of mental health disorders, either past or present; (c) past or present diagnosis of neurological disorders; and (d) non-native Hebrew speakers. We included participants: (a) above 11 years of education; (b) above 18 years of age; and (c) for sighted participants, those with intact vision. All participants provided written informed consent prior to participation in the study, which was approved by the Ethics Committee of Ariel University.

### 2.2. Materials

#### 2.2.1. Deductive Reasoning Tests

(1) Word Context Test [41]. In this task, the meanings of 10 unfamiliar words are deduced based on five clue sentences, which provide some information about the meaning. The clue sentences are presented orally, one at a time, and contain progressively more detailed information. Participants are able to provide an answer after each sentence. If correct, he/she continues to the next word, and, if not, they are presented with another clue sentence that is more specific. The objective of the task was to correctly guess the meaning of the gibberish word using as few clue sentences as possible; the sooner the correct answer is reached, the higher the score, which ranges between 1 and 5 for each presented word. The total score of the test ranges from 0 to 50 with no time limit for this task. The dependent variable is the total correct score in the task. An example item of the task: −Most people need to *prifa* several times a day,−Most people are very careful about their *prifa.*

(2) Deductive reasoning argument task. This test is taken directly from the vocation assessment battery that is used for evaluating cognitive capabilities of applicants for jobs in vocational assessment centers in Israel. Vocational centers in Israel administer the test using a pen and paper for the sighted and orally for people who are blind. In this task, twenty-five reasoning arguments are presented orally. Participants have to choose the correct answer among four possible answers. One point is given for each correct answer, ranging from 0 to 24, without a time limit for the task. The score for this task is the number of correct answers. An example question:

Marcy is Dana’s daughter and Bella’s mother. What is Dana for Bella?

Her mother.Her daughter.Her grandmother.Her granddaughter.

#### 2.2.2. Working Memory Tests

(1) The Digit Span Backward [42]. A task that is specifically used to test the ability to manipulate information [43]. Participants are to repeat a sequence of digits presented orally, in reverse order, beginning with two digits for each sequence length and up to nine digits. If the participants are correct, another single digit is added to the sequence which can reach a total of 16 trials. This procedure is repeated until participants fail two consecutive sequences of the same length. For every correct response, the participants receive one point, within a score range of 0 to 16. The dependent variable for the task is the total number of correct recalls.

(2) Letter–Number Sequencing [42]. In this task, a series of random numbers and letters are presented orally to the participants, who are required to repeat the sequence. First, the numbers in ascending order, and then the letters in alphabetical order. The task is stopped when the participants make an error in three consecutive trials of the same length. A potential total of 21 trials are administered, and 1 point is provided for every correct answer, within a scoring range of 0 to 21. The sum of the correct points is considered as the Letter–Number Sequencing score.

### 2.3. Procedure

Suitable participants according to our exclusion and inclusion criteria received a short explanation of the study, signed an informed consent form (which was read to them in Hebrew), answered the examiners’ demographic questionnaire, and then performed the cognitive tasks. The experiment lasted about 60 min, and the participants were able to ask questions regarding the study after completing the study.

### 2.4. Statistical Analysis

The groups were compared in the demographic measures using Multivariate Analysis of Variance (MANOVA) for age and education and Chi-square analysis for gender. A one-way ANOVA was used to compare the reasoning and working memory scores of the four groups, followed by a Tukey post hoc test to clarify the source of the differences. A non-parametric Kruskal–Wallis H test was also conducted in all comparisons to account for the relatively small sample size. Next, a working memory composite score was created by averaging the two tests after transforming them to Z-scores to explore the extent to which deductive reasoning can be explained by working memory in each group. We combined the scores on their common theoretical and empirical grounds [42] and the pattern of results in the current study. The two reasoning tasks, however, reflect slightly different aspects of reasoning and so were not combined. A regression analysis was used for the working memory composite score of each of the reasoning tasks separately to extract the R-squared scores for the explained variance. The *p* value was adjusted for multiple comparisons of each reasoning task cluster of regressions to *p* = 0.012. We were then able to compare the correlation estimates by using Fisher’s R-to-Z transformation test to assess the differences between the groups. We used SPSS software 25 to analyze the data, and the significance level for all the tests was *p* < 0.05.

## 3. Results

No differences were found between the groups in terms of age and years of education (*F*(3,56) = 1.58, *p* = 0.132), or gender (χ^2^(3) = 0.13, *p* = 0.721).

### 3.1. Reasoning Tasks

(1)Word context task. In the word context test, no significant effect was found between the groups (*F*(3,59) = 1.12, *p* = 0.348, η_p_^2^ = 0.05, H(3) = 3.16, *p* = 0.367; see Figure 1).(2)Deductive reasoning argument task. Comparing the four groups in the deductive reasoning argument task yielded a significant result (*F*(3,59) = 8.18, *p* < 0.001, η_p_^2^ = 0.3, H(3) = 18.39, *p* < 0.001), showing that the sighted participants performed better than the CB group and the blindfolded group, with LB performing better than the blindfolded group.

### 3.2. Memory Tasks

(1) Digit span backwards. A comparison of the groups in the digit span backwards showed a significant effect (*F*(3,59) = 5.58, *p* = 0.002, η_p_^2^ = 0.23, H(3) = 14.99, *p* = 0.002) where CB performed better than the rest of the groups.

(2) Letter–number sequencing task. Similarly, this task was also significant (*F*(3,59) = 4.28, *p* = 0.008, η_p_^2^ = 0.18, H(3) = 10.39, *p* = 0.015), showing again, using the post hoc test, that CB outperformed the other groups (see Figure 2).

### 3.3. Relationship between the Variables

The regression analysis between the working memory composite score and the word context test is not significant in the CB (*F*(1,14) = 1.35; *p* = 0.265), LB (*F*(1,14) = 1.15, *p* = 0.302), blindfolded (*F*(1,14) = 0.42, *p* = 0.527), and sighted participants (*F*(1,14) = 1.57, *p* = 0.232). However, the regression between the working memory composite score and deductive reasoning argument task yields significant results for the CB participants (*F*(1,14) = 10.66, *p* = 0.006), which explains 45% of the variance; for the LB participants (*F*(1,14) = 9.12, *p* = 0.010), it explains 41% of the variance; and for the blindfolded group (*F*(1,14) = 29.26, *p* < 0.001), 69% of the variance. In the sighted group, no association was found (*F*(1,14) = 2.14, *p* = 0.167). Following these results, Fisher’s R-to-Z transformation test showed that none of these correlations were significantly different from each other: CB and LB (*z* = 0.12, *p* = 0.904), CB and blindfolded (*z* = −0.09, *p* = 0.035), and LB and blindfolded (*z* = 1.06, *p* = 0.289), suggesting that performance in the deductive reasoning argument task does not depend on vision or blindness onset (see Figure 3). In addition, testing the differences between the sighted controls and the other groups yielded non-significant results in the CB (*z* = 1.02, *p* = 0.153) and LB groups (*z* = 0.897, *p* = 0.185), but yielded a significant result in the blindfolded group (*z* = 1.95, *p* = 0.025).

## 4. Discussion

Our results show differences in deductive reasoning and working memory in the verbal domain between the sighted, CB, and LB groups. The CB participants outperformed the sighted and LB groups in both of the working memory tasks: the digit span backwards and the letter–number sequencing task. The groups were not different in terms of their deductive reasoning abilities as measured in the word context test. In the deductive reasoning argument task, the blindfolded group performed significantly worse than the LB and sighted participants, while the CB group had lower scores than sighted the participants who were allowed to use pen and paper. Furthermore, we found a link between working memory and deductive reasoning skills in all the groups with blindness, either permanent or temporary and regardless of the level of visual experience. The word context test was not different between the groups. Taken together, our results indicate that visual experience is not necessary for the development of deductive reasoning and working memory skills.

### 4.1. Reasoning Ability

The variability that we found between the groups in terms of reasoning ability is associated with impairment onset. The LB participants, who lost their vision later in life, are not impaired in terms of their reasoning skills as compared to people with normal vision, whereas the CB group did not differ from the blindfolded group and performed worse than the sighted controls. The sensory compensation hypothesis [12] posits that individuals with blindness can function as well as, or better than, sighted individuals due to improvements in their intact senses. Our study has shown that although they lack the ability to see from birth, they did develop reasoning abilities, which are highly related to vision abilities [28]. However, the performance of those with CB was not consistent, showing that different factors other than mere reasoning ability affected their performance. The nature of the task may account for this inconsistency: While both tasks convey deductive abilities, the deductive reasoning argument task relates to relational reasoning, which is considered to be an aspect of deductive reasoning [44]. This ability refers to the inference of relations between variables from other known relations. The deductive reasoning argument task coincides with these features, therefore also entailing a spatial component, which was shown to be more difficult for persons with blindness [28]. The word context test, on the other hand, does not require such inferences of relations, and is thus easier.

Another finding in regards to reasoning is the poor performance of the blindfolded group in the deductive reasoning argument task compared to other groups. It is easy to understand that a group of people who are used to functioning using vision would perform poorly in a situation when vision is absent. It does not seem that the sighted blindfolded group has lower deductive reasoning abilities, however, because they performed equally well as the other groups in the word context test. Rather, working memory load in the deductive reasoning argument task was higher, therefore impairing their ability to perform well. The fact that the sighted controls who used a pen and paper for the task performed better or equally as well as the other groups indicates that it is not reasoning ability but rather working memory load that affects the level of performance. Indeed, the fact that we found that working memory explains 69% of the reasoning variance in the blindfolded group strengthens this conclusion.

### 4.2. Working Memory

The working memory scores of the blindfolded group were not significantly worse than those of the other groups, suggesting that this skill could rely on visual experience to initially develop, but once developed, can also function without access to vision. Our findings further show that the CB group outperformed the LB and sighted participants in those tasks. Many other studies have shown superior working memory abilities in individuals with CB [19,23,31,44] compared to sighted people, and in those with LB compared to those with normal vision [18,45].

Although differences between the sighted groups and the blind groups have been previously reported [46], differences in working memory between the CB and LB groups have not been found. It is possible that the taxing demands of everyday functioning without vision from an early age may help develop compensatory cognitive ability mechanisms for working memory resulting in the better performance of those with CB over the LB population [31,46,47].

Due to the inability to acquire visual information, the use of working memory in everyday life is more frequent among the visually impaired compared to the sighted participants. As a result, the performance of the visually impaired is equal to or better than that of the sighted [18,31]. As such, it is possible to infer that the extended use of working memory, together with the duration of the impairment, affects working memory ability.

### 4.3. Relationship between Visual Experience/Working Memory and Deductive Reasoning Skills

While those with LB did not differ from the sighted participants in the reasoning tasks and also performed worse than those with CB in the working memory tasks, there is still a connection to cognitive skills acquired prior to their impairment [35,40]. Indeed, individuals with late blindness extensively use visual representations for both reasoning [28] and working memory [18,31] in order to maximize their functioning with the sensory information available. These diversified aspects indicate possible qualitative as well as quantitative differences between the CB and LB populations. Consequently, both aspects should be taken into consideration when clinically assessing working memory in the visually impaired population.

The current study further shows that the combined working memory scores of the LB, CB, and blindfolded groups are positively correlated with the deductive reasoning argument task with no difference between the correlations. The sighted participants, however, do not show such an association. This implies that, although there might be differences in working memory and reasoning abilities between visually impaired and sighted people, a moderate extent of the variance in terms of reasoning scores is explained by working memory in the verbal domain [7,8,9,11]. The fact that the sighted participants who used a pen and paper did not show any correlation between the two abilities indicates that working memory load was reduced significantly, and thus deepens our understanding of its role in reasoning ability. This suggestion should be taken with caution, however, because these differences exist only between the sighted controls and blindfolded groups and not between the sighted controls and both CB and LB groups.

Our results are therefore in line with those of others who have suggested that in order to have good reasoning abilities, one must be able to use working memory adequately, which includes both retaining and manipulating data [7,8,9,10,11]. Therefore, the claim of working memory’s involvement in reasoning ability is supported by research in blind and sighted people, also suggesting that the relation between these abilities is not associated with visual experience. Our results further lend support to previous evidence that there are cognitive differences between different etiologies of visual impairment, such that visual experience influences abilities as working memory and reasoning. However, it does not affect their relation, meaning that working memory plays a part in reasoning no matter the level of visual experience. Taken together, the current study does not support the perceptual deficit hypothesis [13], as both clinical groups performed equally as well or better than the sighted controls. Of exception is the deductive reasoning argument task since the experimental conditions differed. On the other hand, our findings do not fully support the sensory compensation hypothesis either [12] since we did not detect superior performance in all the tests or differences between blindness etiologies. Our results seem to support the sensory compensation and brain plasticity hypothesis, which depends on different factors such as etiology of the blindness and the type of abilities measured.

### 4.4. Future Considerations

Considering the importance of providing accommodations for people with visual impairment [47], other cognitive abilities that were beyond the scope of this study, such as inhibition or cognitive flexibility, may also affect reasoning [8] and further explain the performance of those with LB and CB. Future studies designed to specifically explore other executive functions and their impact on reasoning amongst the visually impaired should make use of a wider battery of tests to achieve a better understanding of the abilities implicated in reasoning. In line with this observation, other populations with visual impairments (e.g., short sightedness and those with damaged visual fields) may benefit from similar studies.

## 5. Conclusions

The findings of this study demonstrate that reasoning ability in the verbal domain amongst individuals with visual impairment is not uniform, as different tasks yielded different results. Consequently, it is understood that reasoning is highly associated with several dynamic factors: visual impairment onset, level of reasoning difficulty, and degree of working memory load. Each of these factors has an important impact on the performance of visually impaired individuals’ reasoning function. Therefore, these findings should be taken into consideration when interpreting the reasoning ability of the visually impaired. Assessments carried out in educational and vocational settings should highlight our findings during evaluation so that adjustments and accommodations for visually impaired individuals can be made accordingly.

## Figures and Tables

**Figure 1 sensors-22-02062-f001:**
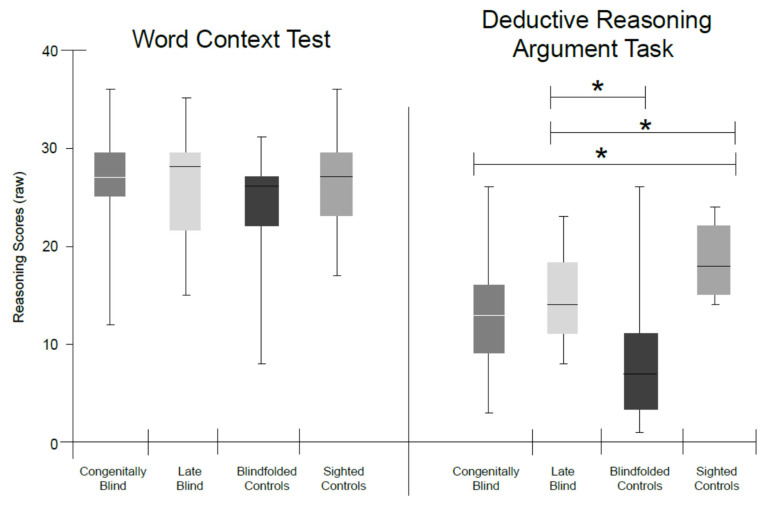
Group comparisons in the reasoning tests scores. Bar graph comparing the performance of congenitally blind, late blind, blindfolded, and sighted controls in the word context test (**left**) and the deductive reasoning argument task (**right**). No difference was found between the groups in the word context test. In the deducting reasoning argument task, the sighted controls performed better than the congenitally blind and blindfolded, and the late blind group performed better than the blindfolded group. * *p* < 0.05.

**Figure 2 sensors-22-02062-f002:**
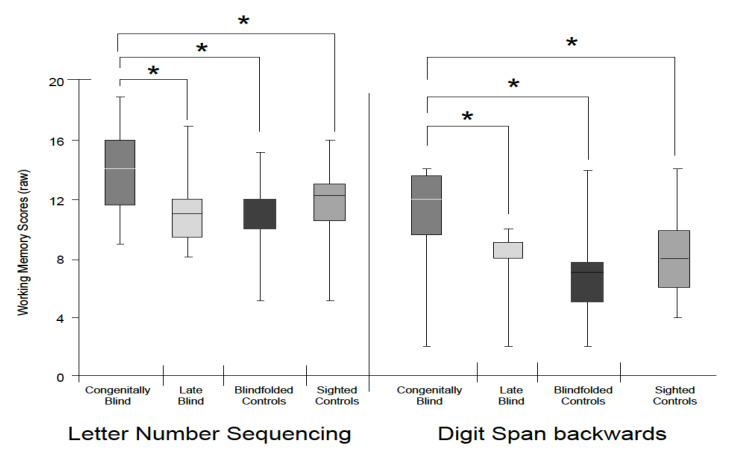
Group comparisons in the working memory tests scores. Bar graph comparing performance of congenitally blind, late blind, blindfolded, and sighted controls in the letter–number sequencing task (**left**) and the digit span backwards task (**right**). The congenital blind group performed better than all the other groups in both working memory tasks. * *p* < 0.05.

**Figure 3 sensors-22-02062-f003:**
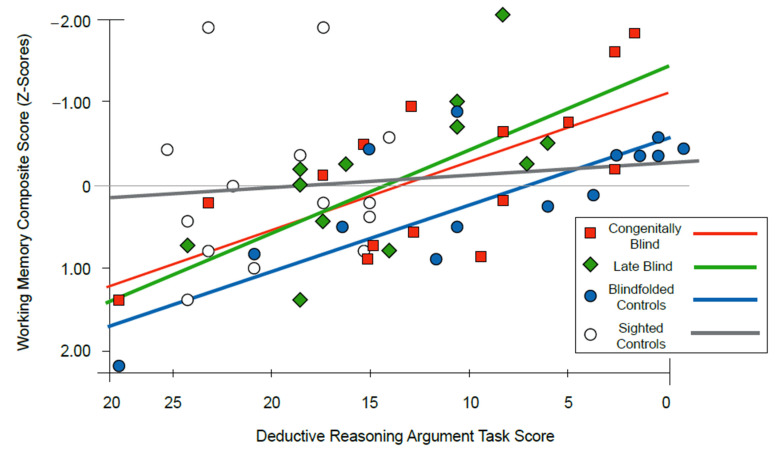
Correlation of reasoning and working memory tasks in the sighted, blindfolded, congenital, and late blindness groups. Scatter plot showing the correlations between deductive reasoning scores (*x*) and working memory scores (*y*) for congenitally blind (red squares), late blind (green lozenges), blindfolded controls (blue circles), and sighted controls (white circles). All the correlations between the deductive reasoning argument task and working memory composite score were significant for all the groups, except for the sighted controls.

**Table 1 sensors-22-02062-t001:** Mean (standard deviation in parenthesis) age and education of participants by group.

	Congenitally Blind	Late Blind	Blindfolded Controls	Sighted Controls
Age	34.4 (5.76)	29.93 (4.36)	26.33 (4.36)	26.33 (4.45)
Education (years)	15.33 (2.09)	13.93 (1.33)	14.93 (2.65)	13.86 (1.55)

## Data Availability

Data are available at: eyalheled@gmail.com.

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
