# Peer review of "Deductive Reasoning and Working Memory Skills in Individuals with Blindness"

_sensors, 2022, doi:10.3390/s22052062_

Round 1

Reviewer 1 Report

Heled et al. present a very interesting study and results on executive abilities and their dependence on vision and visual experience. The design is clever, the statistical methods are adequate, and the conclusions warranted. I have several (mostly clarification) questions to the authors:

Major:

  1. I find it perplexing that the blindfolded group performed so poorly in the deductive reasoning task, although the task had no direct visual component. The advantage of sighted controls, as well as the advantage of late blind, suggests that this task may be aided by visual imagery. However, why would normally-sighted blindfolded participants be impaired in visual imagery? Does the blindfolding confuse them, or do the sighted participants rely on some visual cue (e.g. using their hands or another visual aid to improve performance)? Related to that, the discussion mentions “the sighted participants who use pen and paper” (line 301) – this was not mentioned before – were the sighted participants allowed to use this? If so, this is very important for interpretation and should be highlighted in the methods and discussed.
  2. The observed difference between the two reasoning tasks are very interesting, and suggest that these tests measure different cognitive processes. Specifically, from the example in the deductive reasoning task, it seems that this task specifically involves relational thinking (inferring relations between variables from other known relations), while the word context task does not involve this element that much. In that case, relational thinking might be more reliant on visual imagery than other types of deductive reasoning – which might relate to the general subject of the special issue (spatial thinking). In any case I think this difference between tasks could be discussed a bit more.

Minor:

  1. The regression analyses are not corrected for multiple comparisons, as far as I can tell, although several regressions were performed. (If some of these results are not significant after correction I think that the authors can still describe both the corrected and uncorrected results and I don’t think they need to change any of the conclusions).
  2. Methods – the statistical analysis section is very brief and does not contain all of the methodical details found in the results (e.g. the Fisher R-to-Z transform). Specifically, the regression analysis is not adequately explained (unclear what is regressed from what).
  3. Why were the two working memory tasks combined to a composite score, while the deductive reasoning tasks were not? I assume it is due to the common pattern observed in the working memory tasks, but this should be explicitly described and explained.
  4. Line 165 – difference in education p-value is 1.17 – this must be a mistake.
  5. Lines 219-221 should be removed.

Author Response

Comments and Suggestions for Authors

Heled et al. present a very interesting study and results on executive abilities and their dependence on vision and visual experience. The design is clever, the statistical methods are adequate, and the conclusions warranted. I have several (mostly clarification) questions to the authors:

Thank you for taking the time to revise our manuscript and for your constructive comments that we address here, point by point.

Major:

  1. I find it perplexing that the blindfolded group performed so poorly in the deductive reasoning task, although the task had no direct visual component. The advantage of sighted controls, as well as the advantage of late blind, suggests that this task may be aided by visual imagery. However, why would normally-sighted blindfolded participants be impaired in visual imagery? Does the blindfolding confuse them, or do the sighted participants rely on some visual cue (e.g. using their hands or another visual aid to improve performance)? Related to that, the discussion mentions “the sighted participants who use pen and paper” (line 301) – this was not mentioned before – were the sighted participants allowed to use this? If so, this is very important for interpretation and should be highlighted in the methods and discussed.

 This is an important point. We have made some changes in the text to clarify this issue. Briefly, we think that it is not that these participants are necessarily impaired in visual imagery, but rather that the WM load for the task’s execution influenced performance. Indeed, this group is used to functioning and performing such tasks using vision. Having to perform these tasks while blindfolded disadvantages them. Our results support this suggestion since we  show that:

  1. WM but not reasoning ability (via the Word context test) was found worse in the vision intact groups.
  2. We found significant correlation between WM and deductive reasoning argument task in all blind groups including the blindfolded
  3. The sighted controls (matched by age and education to the blindfolded group) who performed the task visually and used pen and paper had no problem performing the task, indicating that once the WM load was removed, performance was equally well and better than the other groups.

We have added the following two explanations to the text to clarify this point.

In red: (p. 5, lines 95-102), In red:  (p. 8, lines 273-279).

  1. The observed difference between the two reasoning tasks are very interesting, and suggest that these tests measure different cognitive processes. Specifically, from the example in the deductive reasoning task, it seems that this task specifically involves relational thinking (inferring relations between variables from other known relations), while the word context task does not involve this element that much. In that case, relational thinking might be more reliant on visual imagery than other types of deductive reasoning – which might relate to the general subject of the special issue (spatial thinking). In any case I think this difference between tasks could be discussed a bit more.

Indeed, the different reasoning tasks may reflect slightly different aspects of reasoning. We have added the following explanation in the text concerning the differences between the tests. (in red; p. 7, lines 262-269)

Minor:

  1. The regression analyses are not corrected for multiple comparisons, as far as I can tell, although several regressions were performed. (If some of these results are not significant after correction I think that the authors can still describe both the corrected and uncorrected results and I don’t think they need to change any of the conclusions).

 We have added this information in the Statistical analysis section in red (p. 4, lines 175-176).

  1. Methods – the statistical analysis section is very brief and does not contain all of the methodical details found in the results (e.g. the Fisher R-to-Z transform). Specifically, the regression analysis is not adequately explained (unclear what is regressed from what).

 We have adjusted the Statistical section to include all the relevant information: (in red; p. 4, lines 168-178)  

  1. Why were the two working memory tasks combined to a composite score, while the deductive reasoning tasks were not? I assume it is due to the common pattern observed in the working memory tasks, but this should be explicitly described and explained.

This point is related to the previous point (#2) raised by the reviewer concerning the differences between the tests. From a theoretical point of view, the two working memory tests reflect the same skills, while the reasoning tests reflect slightly different aspects of reasoning. This is well established in the literature since the working memory tests are part of the WAIS-III battery (Wechsler, 1997) that show they are loaded on the same factor. Furthermore, statistically, our results confirm this as well (see previous point). This is not the case for deductive reasoning, both empirically and in terms of our results. For these reasons we combined the scores for the working memory test and not for the deductive reasoning task. This point is further clarified in the text in red. (p. 4 lines, 168-178)  

  1. Line 165 – difference in education p-value is 1.17 – this must be a mistake.

We thank the reviewer for this note, as following it we double checked the analysis and corrected it to MANOVA as there are two comparisons here (age and education). We corrected the report accordingly. “No differences are found between the groups in terms of age and education (F(3,56) = 1.58, p = .132), or gender (χ2(3) = .13, p = .721).” (p. 4, lines 181-182)

  1. Lines 219-221 should be removed.

The sentence was removed from the text.

Reviewer 2 Report

Summary: This manuscript reports an assessment of the working memory and reasoning abilities of groups of congenitally blind, late-onset blind, sighted, and sighted blindfolded adults. Data show: a consistent advantage for congenitally blind adults on two tests of working memory; no differences in word context reasoning but some differences in deductive reasoning; and a positive relationship between composite WM performance and deductive reasoning performance in CB, LB, and sighted blindfolded controls but not in sighted controls. These results are put in context with several other studies showing conflicting evidence for advantages in cognitive capacities of blind individuals.

I found this paper very clearly organized and written, on an interesting and important topic of great relevance to the focus of this special issue, and with generally strong methods and interpretation of results. I have what turned out to be quite a few points of suggestion, but I expect these can be addressed in a major revision. I am positively disposed towards this paper and would be glad to recommend publication if these points can be addressed.

Major points:

  1. Deductive reasoning argument task: Is this a task of your own design? I’d like to know more about how this was selected, and what the logic was behind including this test. This is particularly important because results diverge from the more well-established word context task, for which you report no differences across vision condition.
  2. I’m not sure it’s appropriate to form a composite working memory score to regress on each reasoning score independently. Is there a reason not to either (1) treat all four scores as separate and do pairwise regressions or (2) regress a composite WM score on a composite reasoning score, with the assumption that both reasoning tests are measures of the same underlying construct as, presumably, are both WM scores? It would be helpful to know the logic behind these decisions and if/why the authors feel these tests are the most appropriate ones to assess relationships between WM and reasoning in these data.
    1. Note that I would also be less concerned about this choice if it was made for principled reasons prior to any data analysis, rather than after exploring potential relationships between multiple dependent variables/composite scores.
  3. Line 214: Is the null correlation in sighted controls significantly different from any or all of the three significant groups? This is important for assessing whether there are reliable and interpretable differences in sighted controls, or perhaps just too much variability in the regression for that group to make strong assessments. Without this difference, saying that there is no relationship in sighted controls (line 296) is not statistically justified, as the lack of a significant effect is not evidence for the absence of that effect. This being said, I think your suggestion that this difference could be explained (if statistically reliable) by a decrease in WM load through pen-and-paper use in sighted controls is a good suggestion.
  4. General discussion: I recommend coming back to the two theories you outline in the introduction to help the reader follow what to make, overall, of your data. Do your data support the sensory compensation hypothesis, the perceptual deficit hypothesis, or neither?

Minor points:

  1. Line 34: should this read “two of…” or is there an ability missing?
  2. Paragraph beginning line 60: It would be nice to have reasons spelled out for why differences might be expected between congenital and late blindness’ effects on WM and reasoning. Is this mainly because of differences in brain plasticity in the two groups, and therefore mainly expected in the ‘sensory compensation hypothesis’ account but not the deficit account, or something else?
  3. Line 76: “vision is an important, yet, non-crucial factor associated with reasoning.” The meaning of this is unclear, could you be more precise? In particular, I’m not sure what it could mean for vision to be crucial to reasoning.
  4. Line 77: “there are differences in the performance of sighted, CB 77 and LB people for deductive reasoning and working memory skills”
  5. Line 91: I assume this average age of blindness onset is only for the LB participants, and isn’t computing average onset for both CB and LB together? Please specify this.
  6. Line 145: It would be good to know what these exclusion/inclusion criteria are. Could you explain this in more detail, at least in your response to reviewers but potentially (briefly) in the manuscript as well?
  7. Line 167: It’s odd to include this table but then not reference it or even the normality tests at all. Please add (briefly) details about these tests and what corrections, if any, were made based on them.
  8. Results: Were all pairwise comparisons not mentioned (e.g., the difference between sighted controls and LB participants on the deductive non-significant?
  9. Line 210: What is being regressed against what in the results on deductive reasoning arguments? I think this is the relationship between deductive reasoning and working memory composite score, but it would be good to specify.
  10. Line 219: I think this is an instructions paragraph that should now be removed?
  11. I would like to see a figure of data of the relationship between working memory composite score and the word context task too, at least in the responses to reviewers and/or a supplement.
  12. Line 234: It would help the reader to specify that these two tasks (digit span and letter-number sequencing) are both WM tasks, since otherwise the start of the sentence makes it sounds like they represent both reasoning and WM.
  13. Line 239: don’t you find no such relationship for sighted controls (with no blindfold)?
  14. Line 247: ‘Did not outperform’ sounds like there was no difference between CB and the sighted/sighted-blindfolded groups, rather than there being a significantly lower score for CB for the sighted group.
  15. Line 248-252: This reads like you are trying to have it both ways, with your results both partially supporting better function in CB (with compensatory strategies) and that performance depends on early vision (which would predict a deficit). Am I right in thinking that the latter of these is more in line with your data? You don’t report any direct evidence for compensatory strategies, and performance is similar to worse on reasoning in CB participants.
  16. Line 257: this is a good point, but it also makes me wonder to what extent this deductive reasoning task itself relies on working memory rather than/in addition to reasoning. This is related to my major point 1 – could you comment on the extent to which this task is partially a WM task, if at all? (This could also explain the relatively large R^2 values for the deductive reasoning-WM regression).
  17. Line 261: I’m not sure that intact WM in sighted blindfolded controls means WM develops independently of visual experience – WM could rely on visual experience to initially develop but, once developed, function without access to vision.
  18. Paragraph beginning line 271: How do your results relate to this suggestion? This paragraph feels out of place without some connection to the present work.

Author Response

REVIEWER 2

 Comments and Suggestions for Authors

 Summary: This manuscript reports an assessment of the working memory and reasoning abilities of groups of congenitally blind, late-onset blind, sighted, and sighted blindfolded adults. Data show: a consistent advantage for congenitally blind adults on two tests of working memory; no differences in word context reasoning but some differences in deductive reasoning; and a positive relationship between composite WM performance and deductive reasoning performance in CB, LB, and sighted blindfolded controls but not in sighted controls. These results are put in context with several other studies showing conflicting evidence for advantages in cognitive capacities of blind individuals.

I found this paper very clearly organized and written, on an interesting and important topic of great relevance to the focus of this special issue, and with generally strong methods and interpretation of results. I have what turned out to be quite a few points of suggestion, but I expect these can be addressed in a major revision. I am positively disposed towards this paper and would be glad to recommend publication if these points can be addressed.

We would like to thank the reviewer for their careful review of our manuscript, and for kind and constructive comments. We answer each of your queries below explaining the changes we made to the text:

Major points:

  1. Deductive reasoning argument task: Is this a task of your own design? I’d like to know more about how this was selected, and what the logic was behind including this test. This is particularly important because results diverge from the more well-established word context task, for which you report no differences across vision condition.

Our study is part of a larger study that deals with vocational aspects of people with blindness and their chances of receiving equal opportunity to sighted individuals in vocational evaluation centers in Israel. The study explores if the conditions that are given to those with blindness are equal to those without it. One of the aspects that were evaluated was the cognitive capabilities section in which cognitive abilities are assessed through various tests, while one of them is a reasoning task. The deductive reasoning argument task was taken from the assessment battery used in these centers, while the WM tasks are not part of the battery and we wanted to explore if WM as an important ability that influences reasoning, is the same in both groups. That is why we chose tests that were shown to be related theoretically and empirically (taken from the WAIS-III). This aspect was mentioned in the end of the introduction as a potential contribution of the study ("Psychologists and other professionals in the field of cognitive assessment in educational and vocational settings could consider these important aspect…"). We have elaborated on this idea and added explanations about the test in the Methods section. In red: (p. 3, lines 123-127).

Regarding the second part of the comment, we have added further explanations in the discussion section that concern the deductive reasoning argument task’s nature compared to the word context test.  In red: (p. 8, lines 261-269)   

  1. I’m not sure it’s appropriate to form a composite working memory score to regress on each reasoning score independently. Is there a reason not to either (1) treat all four scores as separate and do pairwise regressions or (2) regress a composite WM score on a composite reasoning score, with the assumption that both reasoning tests are measures of the same underlying construct as, presumably, are both WM scores? It would be helpful to know the logic behind these decisions and if/why the authors feel these tests are the most appropriate ones to assess relationships between WM and reasoning in these data.
    1. Note that I would also be less concerned about this choice if it was made for principled reasons prior to any data analysis, rather than after exploring potential relationships between multiple dependent variables/composite scores.

Yes, this is a deliberate choice taken on the grounds of empirical and theoretical premises prior to data analysis: The two working memory tests were chosen because they have similar theoretical and empirical basis. From a theoretical point of view, the two working memory tests reflect the same skills, while the reasoning tests reflect slightly different aspects of reasoning. This is well established in the literature since the working memory tests are part of the WAIS-III battery (Wechsler, 1997) that show they are loaded on the same factor. The link between the two reasoning tasks is not well established empirically, and our findings also seem to reflect that these tests reflect slightly different skills. Furthermore, statistically, our results confirm this as well. This is not the case for deductive reasoning, both empirically and in terms of our results. Furthermore, for reasons of parsimony it was suggested to us by a statistics consultant to the study (Prof. Bellavance, University of Montreal) to combine the score of the two tests. We have added an explanation in the data analysis section: In red: (p. 4 lines 170-172)   

  1. Line 214: Is the null correlation in sighted controls significantly different from any or all of the three significant groups? This is important for assessing whether there are reliable and interpretable differences in sighted controls, or perhaps just too much variability in the regression for that group to make strong assessments. Without this difference, saying that there is no relationship in sighted controls (line 296) is not statistically justified, as the lack of a significant effect is not evidence for the absence of that effect. This being said, I think your suggestion that this difference could be explained (if statistically reliable) by a decrease in WM load through pen-and-paper use in sighted controls is a good suggestion.

After testing the differences in correlation estimates between the sighted controls and the other groups, we find significant results only between the control and blindfolded groups but not in the CB and LB groups. We have clarified this issue in red the text:  (p. 9, lines 316-318)

  1. General discussion: I recommend coming back to the two theories you outline in the introduction to help the reader follow what to make, overall, of your data. Do your data support the sensory compensation hypothesis, the perceptual deficit hypothesis, or neither?

We added the following explanations to the Discussion:

In red in the text (p. 9, line 328-335)  

Minor points:

  1. Line 34: should this read “two of…” or is there an ability missing?

We thank the reviewer for this comment. We corrected the wording to two instead of three.

  1. Paragraph beginning line 60: It would be nice to have reasons spelled out for why differences might be expected between congenital and late blindness’ effects on WM and reasoning. Is this mainly because of differences in brain plasticity in the two groups, and therefore mainly expected in the ‘sensory compensation hypothesis’ account but not the deficit account, or something else?

We have added the following sentences to the text in red to clarify this issue:  (p. 2, lines 65-69)

  1. Line 76: “vision is an important, yet, non-crucial factor associated with reasoning.” The meaning of this is unclear, could you be more precise? In particular, I’m not sure what it could mean for vision to be crucial to reasoning.

We have corrected the wording as follows: “Our general findings show that vision is not essential for the development of deductive reasoning and working memory skills”. (in red: p. 2, lines 80-81)

  1. Line 77: “there are differences in the performance of sighted, CB 77 and LB people for deductive reasoning and working memory skills”

We have removed the sentence.

  1. Line 91: I assume this average age of blindness onset is only for the LB participants, and isn’t computing average onset for both CB and LB together? Please specify this.

Yes, CB participants are all blind from birth so were not included in this analysis. The text was corrected to reflect that the average age of blindness onset includes only the LB group:  “The average age of blindness onset in the LB group is 19.66.” (in red, p. 2, line 93-94)

  1. Line 145: It would be good to know what these exclusion/inclusion criteria are. Could you explain this in more detail, at least in your response to reviewers but potentially (briefly) in the manuscript as well?

We have added this information for all participants:  In red: (p. 2, lines 98-103).

  1. Line 167: It’s odd to include this table but then not reference it or even the normality tests at all. Please add (briefly) details about these tests and what corrections, if any, were made based on them.

The table was removed.

  1. Results: Were all pairwise comparisons not mentioned (e.g., the difference between sighted controls and LB participants on the deductive non-significant?

Once the general model of the comparison is non-significant, we cannot further test for pairwise comparisons. However, following the reviewer question we checked them and found that all were non-significant.  

  1. Line 210: What is being regressed against what in the results on deductive reasoning arguments? I think this is the relationship between deductive reasoning and working memory composite score, but it would be good to specify.

We added this information as requested:  In red: (p. 6, line 219)

  1. Line 219: I think this is an instructions paragraph that should now be removed?

The paragraph was removed.

  1. I would like to see a figure of data of the relationship between working memory composite score and the word context task too, at least in the responses to reviewers and/or a supplement.

We have created the following graph (please see the word document,in attachment) showing the relationship between the working memory composite score and the word context task. Although this is an interesting relationship, since it is not significant, we believe it would not enhance our manuscript’s impact if included in the published article.

  1. Line 234: It would help the reader to specify that these two tasks (digit span and letter-number sequencing) are both WM tasks, since otherwise the start of the sentence makes it sounds like they represent both reasoning and WM.
  • We have added the following sentence:

”CB participants outperform the sighted and LB groups in both working memory tasks - the digit span backward and the letter-number sequencing task.” (p. 6, lines 242-243)

  1. Line 239: don’t you find no such relationship for sighted controls (with no blindfold)?
  • We adjusted the text accordingly: “in all groups with blindness either permanent or temporary”. (p. 7, line 248)
  1. Line 247: ‘Did not outperform’ sounds like there was no difference between CB and the sighted/sighted-blindfolded groups, rather than there being a significantly lower score for CB for the sighted group.
  • We adjusted the sentence to clarify this issue: “…did not differ from blindfolded group and performed worse than sighted controls.” (p. 7, line 256-257)
  1. Line 248-252: This reads like you are trying to have it both ways, with your results both partially supporting better function in CB (with compensatory strategies) and that performance depends on early vision (which would predict a deficit). Am I right in thinking that the latter of these is more in line with your data? You don’t report any direct evidence for compensatory strategies, and performance is similar to worse on reasoning in CB participants.
  • Yes, you are correct. We have added a sentence clarifying our position on this issue stating that reasoning ability can indeed be improved due to its practice in intact senses, as the compensatory hypothesis suggests:

”The sensory compensation hypothesis [12] posits that individuals with blindness can function as well as and better than sighted individuals due to improving their intact senses. Our study have shown that although they lack the ability to see from birth they did develop reasoning abilities, which are highly related to vision abilities [28]. However, CBs function was not consistent, showing that reasons other than mere reasoning ability affected their function. The nature of the task may account for this inconsistency: While both tasks convey deductive abilities the deductive reasoning argument task relates to relational reasoning, which is considered to be an aspect of deductive reasoning [43]. This ability refers to the inference of relations between variables from other known relations. The deductive reasoning argument task coincides with these features, therefore entails a spatial component as well, which was shown to be more difficult for persons with blindness [28]. The word context test on the other hand does not require such inferences on relations, thus easier.” (p. 7, lines 257-269)

  1. Line 257: this is a good point, but it also makes me wonder to what extent this deductive reasoning task itself relies on working memory rather than/in addition to reasoning. This is related to my major point 1 – could you comment on the extent to which this task is partially a WM task, if at all? (This could also explain the relatively large R^2 values for the deductive reasoning-WM regression).

In line with the reviewer’s major point #1, we mention that part of the performance in the deductive reasoning argument task relies on WM, which is also true for the word context test but to a far lesser extent. We have added the following sentence in the text:

“It does not seem that the sighted blindfolded group has lower deductive reasoning abilities however, because they performed equally well as the other groups in the word context test. Rather, working memory load in this task is higher therefore impairing their ability to perform well in this task. The fact sighted controls who use pen and paper for the task perform better or equally well to the other groups, indicates it is not reasoning ability but rather working memory load that affects the level of performance. Indeed, the fact that we find working memory explains 69% of the reasoning variance in the blindfolded group, strengthens this conclusion. ‘’ (p. 8, lines 273-280).  

  1. Line 261: I’m not sure that intact WM in sighted blindfolded controls means WM develops independently of visual experience – WM could rely on visual experience to initially develop but, once developed, function without access to vision.
  • We strongly agree with this point raised by the reviewer and have changed the text accordingly:

“The working memory scores of the blindfolded group is not significantly worse than the other groups, suggesting that this skill could rely on visual experience to initially develop but once developed, function without access to vision”. (p. 8, lines 284-285)

  1. Paragraph beginning line 271: How do your results relate to this suggestion? This paragraph feels out of place without some connection to the present work.

Round 2

Reviewer 2 Report

The authors did an excellent job revising this manuscript. I have no further comments and recommend publication.